

# Comprehensive analysis reveals a metabolic ten-gene signature in hepatocellular carcinoma

Zhipeng Zhu[1],*, Lulu Li[1],*, Jiuhua Xu[2], Weipeng Ye[2], Borong Chen[1], Junjie Zeng[1] and Zhengjie Huang[1,2]

[1] Department of Gastrointestinal Surgery, Xiamen Cancer Center, The First Affiliated Hospital of Xiamen University, Xiamen, Fujian, China
[2] Department of Clinical Medicine, Fujian Medical University, Xiamen, Fujian, China
* These authors contributed equally to this work.

Corresponding author
Zhengjie Huang,
huangzhengjie@xmu.edu.cn

## ABSTRACT

**Background:** Due to the complicated molecular and cellular heterogeneity in hepatocellular carcinoma (HCC), the morbidity and mortality still remains high level in the world. However, the number of novel metabolic biomarkers and prognostic models could be applied to predict the survival of HCC patients is still small. In this study, we constructed a metabolic gene signature by systematically analyzing the data from The Cancer Genome Atlas (TCGA), Gene Expression Omnibus (GEO) and International Cancer Genome Consortium (ICGC).
**Methods:** Differentially expressed genes (DEGs) between tumors and paired non-tumor samples of 50 patients from TCGA dataset were calculated for subsequent analysis. Univariate cox proportional hazard regression and LASSO analysis were performed to construct a gene signature. The Kaplan–Meier analysis, time-dependent receiver operating characteristic (ROC), Univariate and Multivariate Cox regression analysis, stratification analysis were used to assess the prognostic value of the gene signature. Furthermore, the reliability and validity were validated in four types of testing cohorts. Moreover, the diagnostic capability of the gene signature was investigated to further explore the clinical significance. Finally, Go enrichment analysis and Gene Set Enrichment Analysis (GSEA) have been performed to reveal the different biological processes and signaling pathways which were active in high risk or low risk group.
**Results:** Ten prognostic genes were identified and a gene signature were constructed to predict overall survival (OS). The gene signature has demonstrated an excellent ability for predicting survival prognosis. Univariate and Multivariate analysis revealed the gene signature was an independent prognostic factor. Furthermore, stratification analysis indicated the model was a clinically and statistically significant for all subgroups. Moreover, the gene signature demonstrated a high diagnostic capability in differentiating normal tissue and HCC. Finally, several significant biological processes and pathways have been identified to provide new insights into the development of HCC.
**Conclusion:** The study have identified ten metabolic prognostic genes and developed a prognostic gene signature to provide more powerful prognostic information and improve the survival prediction for HCC.

# INTRODUCTION

Primary liver cancer is the seventh most commonly occurring cancer in 2018, and the second most common cause of cancer mortality worldwide. The overall 5-year survival of patients with liver cancer is currently 10–20%. Among them, HCC accounts for most of the primary liver cancer (75–85%), which is characterized by high invasiveness, high metastasis potential and low survival rate (*Bray et al., 2018*; *Yu et al., 2017*). The situation is even more serious in China; liver cancer has a new incidence of 370,000 in 2015, ranking fourth in the number of malignant tumors, 326,000 deaths, and second in the number of deaths (*Zheng et al., 2019*). However, there is still a lack of effective biomarkers for prediction of high recurrence populations, death risk and target therapies. Thus, identification of effective biomarker for the prognosis of HCC is urgent for the diagnosis and treatment of HCC.

Traditional serum markers have been proved as potential tumor markers for prognostic in HCC, such as alpha-fetoprotein (AFP) (*Hanazaki et al., 2001*). However, AFP is only elevated in about half of the HCC patients and significant tumor burden limits its usefulness in screening and operable therapy (*Tangkijvanich et al., 2000*). C-reactive protein (CRP) and Platelet lymphocyte ratio (PLR) could be considered as tumor markers for low-AFP HCC patients (*Suner et al., 2019a*), and possess parameter values for tumor growth and invasiveness (*Suner et al., 2019b*). However, the effects of CRP or PLR on survival reveal unclear. What's more, traditional prognosis markers for HCC only focused on single biomarker, including enzymes and isoenzymes, growth factors and their receptors, tumor-associated antigens, microRNAs (miRNAs) and long noncoding RNAs (lncRNAs) (*Mann et al., 2007*; *Singhal et al., 2012*; *Yu et al., 2007*; *Yu, Chen & Ding, 2010*), which may lack sensitivity and specificity. With the development of high-throughput technologies, many new potential biomarkers are easier and the gene prognostic signature is more likely to generated for prognosis in HCC.

Altered cellular metabolism plays a key role for cancerous cells, and cancerous cell metabolism reprograming is considered the novel hallmark of cancer in the future (*Hanahan & Weinberg, 2011*). Previous studies have indicated that metabolism alteration could promote cell proliferation and progression, and XR, *Xu et al. (2001)* shown that the transcription level of metabolic genes has changed in HCC. Thus several metabolism-related genes may play a role in the occurrence and development of HCC. However, the number of novel metabolic biomarkers and prognostic models could be applied to predict the survival of HCC patients is still small. *Jiang et al. (2019)* have constructed a glycolysis gene prognostic signature of HCC, but only one validation cohort have been used to prove the performance of the predicted model, and it lacks of comparison of performance with other different biomarker. *Benfeitas et al. (2019)* built a four-gene survival signature of HCC; systematic analysis is required to further prove the predicted value. Inspired by all these works, our research combined with clinically significant

**Table 1 Summary of patient demographics and clinical characteristics.**

| Characteristic | Training cohort | Internal testing cohort | GSE14520 testing cohort | ICGC testing cohort | Entire testing cohort |
|---|---|---|---|---|---|
| Gender | | | | | |
| Male | 246 (67%) | 121 (66%) | 191 (87%) | 182 (75%) | 619 (75%) |
| Female | 119 (33%) | 63 (34%) | 29 (13%) | 61 (25%) | 209 (25%) |
| Age | | | | | |
| <60 | 165 (45%) | 83 (45%) | 177 (80%) | 45 (18%) | 387 (47%) |
| >=60 | 200 (55%) | 101 (55%) | 43 (20%) | 198 (82%) | 441 (53%) |
| Stage | | | | | |
| Stage I–II | 254 (74%) | 136 (78%) | 170 (77%) | 146 (60%) | 570 (70%) |
| Stage III–IV | 88 (26%) | 38 (22%) | 50 (23%) | 97 (40%) | 235 (30%) |
| Vital status | | | | | |
| Living | 239 (65%) | 125 (68%) | 135 (61%) | 199 (82%) | 573 (69%) |
| Dead | 126 (35%) | 59 (32%) | 85 (39%) | 45 (18%) | 255 (31%) |

metabolic genes to make a gene prognosis model, which could provide better guidance for the survival and prognosis of HCC.

# METHOD

## Identification and acquisition of TCGC, GEO and ICGC data

Gene expression profiles and clinical data associating with HCC were identified and acquired from TCGA project (https://cancergenome.nih.gov/), GEO database (https://www.ncbi.nlm.nih.gov/gds/) and ICGC project (https://icgc.org/). A total of 415 candidates (365 patients and 50 normal candidates) and the clinical information (age, gender, grade, stage, myometrial invasion, lymph node status, distant metastasis status) were identified and acquired from TCGA dataset. A total of 220 patients and the clinical information (age, gender, ALT, AFP, stage) were identified and acquired from GEO database. A total of 243 patients and the clinical information (gender, age, stage, prior Malignancy) were identified and acquired from ICGC dataset.

## Patients characteristics and grouping

A total of 365 patients from TCGA dataset, used as the training cohort. Using the R package Caret, with the ratio of 1:1 in a random manner, 184 patients from TCGA dataset were selected as the internal testing cohort. A total of 243 patients from ICGC dataset and 220 patients from GSE14520 dataset, used as the external testing cohorts. Finally, we integrated all the 828 patients from the TCGA, GEO and ICGC datasets to use as the entire testing cohort. The clinical information of the four cohorts is summarized in Table 1.

## Identification of metabolism related genes

Metabolism-related genes in the KEGG pathway associated with metabolism were screened from the GSEA (http://software.broadinstitute.org/gsea/index.jsp), and the

overlapping metabolism-related genes were identified from TCGA, GSE14520, and ICGC gene expression profile.

## Prognostic genes were identified and gene signature were constructed by utilizing training cohort

Using the Limma version 3.36.2 R package, DEGs were calculated between tumors and paired non-tumor samples of 50 patients from TCGA dataset, the adjusted $P$-value < 0.05 and absolute log2 fold change (FC) > 1.5 were considered as the selection criterion. By using the R package *survival* and *Coxph* function, Univariate Cox proportional hazard regression analysis was performed to discover the prognostic genes in the training cohort, with the adjusted $P$-value < 0.05 as the significance cutoff. We further narrowed the gene range to construct a gene signature by performing LASSO analysis, R package *glmnet* was used to perform LASSO analysis.

## The performance of gene signature

With the risk-formula: Risk score = expression of gene1 × β1gene1 + expression of gene2 × β2gene2 +…… expression of genen × βngenen, patients in training cohort were divided into low or high risk group basing on the median risk score. The expression level was compared between low and high risk group. Kaplan–Meier analysis was performed to compare the survival rate between the two groups. ROC curve analysis for OS was performed to assess the clinically predictive ability of the gene signature. Next, Univariate and Multivariate Cox proportional hazards analysis were performed to investigate whether the gene signature could be independent of other clinical parameters, including age, gender, grade, stage, myometrial invasion, lymph node status, distant metastasis status. Furthermore, stratification analysis were used to assess the prognostic value of the gene signature in different subgroups stratified clinical variables. Moreover, the diagnostic capability of the gene signature was investigated to further explore the clinical significance, including differentiating normal tissue and HCC, different stages and grades.

## Validation of the gene signature

Internal testing cohort, GSE14520 testing cohort, ICGC testing cohort and entire testing cohort were used to validate the reliability and validity of the gene signature. According to the risk-formula and the median risk score, the patients from the four testing cohorts were divided into low or high risk group. And the same analyses were performed to validate the performance, including Kaplan–Meier analysis, the ROC curve analysis, Univariate and Multivariate Cox proportional hazards analysis, stratification analysis.

## Go enrichment analysis and Gene set enrichment analysis

Metabolism related DEGs were identified between high- and low-risk groups, with corrected $P$-value < 0.05 and absolute log fold change (FC) > 1.5 being considered as the cutoff criterion. Next, gene ontology processes were considered as enriched by using the DAVID database (https://david.ncifcrf.gov/). Furthermore, we generated an ordered list of all genes according to their correlation with two subtypes and elucidated the significant survival difference between high- and low-risk groups by GSEA. A total of 1,000 times

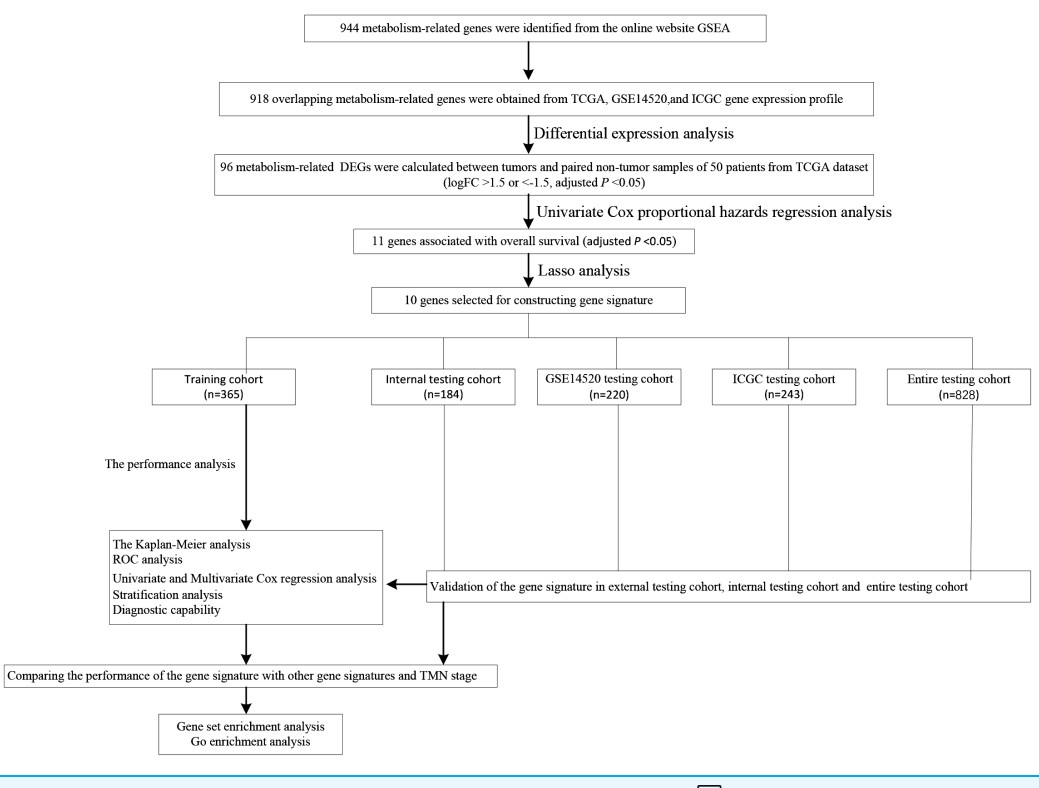

944 metabolism-related genes were identified from the online website GSEA

↓

918 overlapping metabolism-related genes were obtained from TCGA, GSE14520,and ICGC gene expression profile

Differential expression analysis

96 metabolism-related DEGs were calculated between tumors and paired non-tumor samples of 50 patients from TCGA dataset
(logFC >1.5 or <-1.5, adjusted *P* <0.05)

Univariate Cox proportional hazards regression analysis

11 genes associated with overall survival (adjusted *P* <0.05)

Lasso analysis

10 genes selected for constructing gene signature

| Training cohort (n=365) | Internal testing cohort (n=184) | GSE14520 testing cohort (n=220) | ICGC testing cohort (n=243) | Entire testing cohort (n=828) |

The performance analysis

The Kaplan-Meier analysis
ROC analysis
Univariate and Multivariate Cox regression analysis
Stratification analysis
Diagnostic capability

Validation of the gene signature in external testing cohort, internal testing cohort and entire testing cohort

Comparing the performance of the gene signature with other gene signatures and TMN stage

Gene set enrichment analysis
Go enrichment analysis

**Figure 1 The schematic workflow of the study.**

were performed for gene set permutations. The nominal *P* value was used to sort the pathways enriched in each phenotype.

## Statistical analysis

*P* < 0.05 was considered statistically significant, The statistical analyses were conducted by employing the R (version 3.4.3) and GraphPad Prism 7.

## RESULT

### Identification of prognostic genes

We carry on our study as described in the flow chart (Fig. 1). A total of 944 metabolism-related genes were identified from the online website GSEA (Table S1) and 918 overlapping metabolism-related genes were obtained between TCGA, GSE14520 and ICGC gene expression profile (Table S2). A total of 71 up-regulated genes and 25 down-regulated genes were calculated for subsequent analysis (Fig. 2; Table S3). By conducting Univariate Cox proportional hazards regression analysis, 11 significant genes associated with OS were obtained for further analysis (adjusted *P* < 0.05) (Table S4).

### Establishment of gene signature from the training cohort

LASSO analysis was performed to narrow the gene range to construct a ten-gene signature from 11 significant genes (Risk-formula: Risk score = expression of G6PD $\times$ 0.0015558631423743 + expression of LPCAT1 $\times$ 0.00310356967612766 + expression of ME1 $\times$ 0.00632401235412166 + expression of PRIM1 $\times$ 0.0026285256475841 + expression

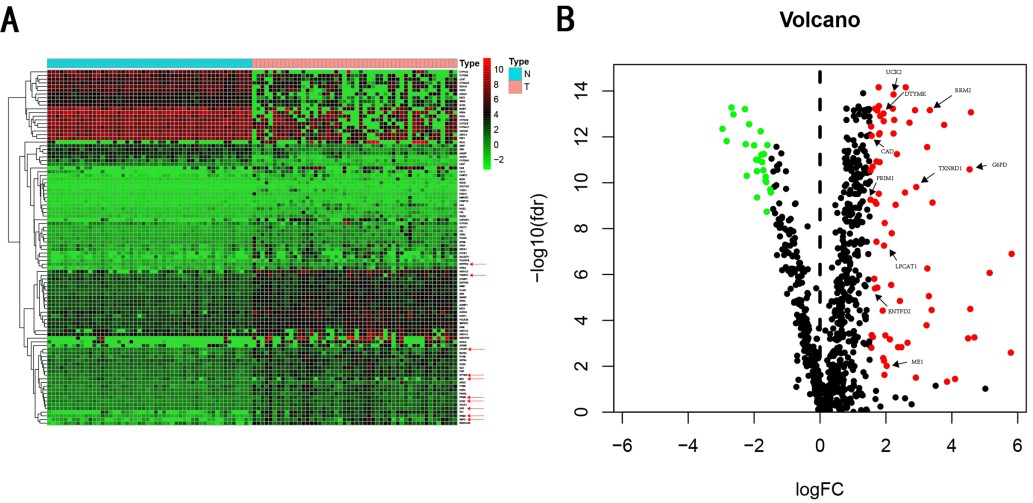

**Figure 2 Heatmap and Volcano plot of metabolism-related DEGs.** (A) Heatmap of metabolism-related DEGs. Red indicates that the gene expression is relatively high, green indicates that the gene expression is relatively low, and white indicates no significant changes in gene expression (FDR < 0.05, absolute log FC > 1.5). A total of 10 prognostic genes were marked using red arrow. (B) Volcano plot of metabolism-related DEGs. The red points represent high expression genes, the green points represent low expression genes, the black points represent genes with no significant difference (FDR < 0.05, absolute log FC > 1.5). A total of 10 prognostic genes were marked using black arrow.

of RRM2 × 0.00979851158278189 + expression of TXNRD1 × 0.00783495109084 + expression of UCK2 × 0.057281553558209 + expression of CAD × 0.06164749118803 + expression of DTYMK × 0.00849922964649245 + expression of ENTPD2 × 0.03691738437542). The risk score was computed for each patient in the training cohort, 365 patients from training cohort were divided into low risk group (183 patients) and high risk group (182 patients) according to the median risk score: 0.73. A total of 243 patients from ICGC testing cohort were divided into low risk group (100 patients) and high risk group (143 patients). A total of 220 patients in GSE14520 testing cohort were divided into low risk group (112 patients) and high group (108 patients). A total of 184 patients in the internal testing cohort were divided into low risk group (95 patients) and high risk group (89 patients), 828 patients in the entire testing cohort were divided into low risk group (356 patients) and high risk group (432 patients). The 10 prognostic genes expression level distribution between low and high risk group of the training cohort was showed in Fig. 3A, the expression of ten prognostic genes in high risk group was higher than in low risk group, the result was consisted with internal testing cohort in Fig. 3B, GSE14520 testing cohort in Fig. 3C and ICGC testing cohort in Fig. 3D.

## The performance of gene signature

As showed in Fig. 4A, with the increasing risk score, patients in the training cohort have a worse OS, the expression ten prognostic genes increased. The ROC curve was presented to assess the clinically predictive ability of the gene signature, the AUC for 1-year (Fig. 4B), 3-year (Fig. 4C), and 5-year (Fig. 4D) OS were 0.805, 0.756, 0.716 for

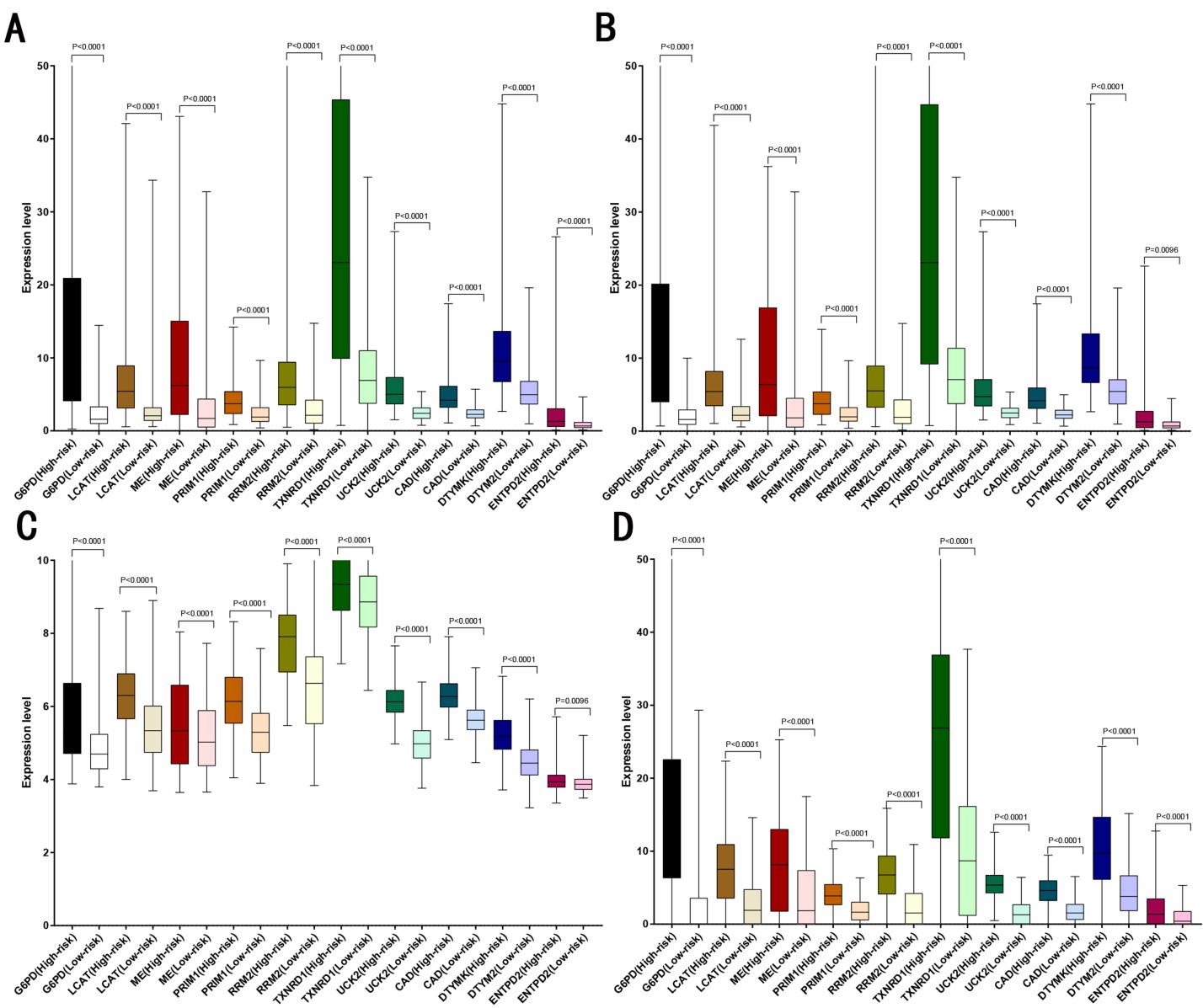

**Figure 3** Expression of the ten genes in low- and high-risk groups of training cohort, internal testing cohort, GSE14520 testing cohort and ICGC testing cohort. (A) Training cohort. (B) Internal testing cohort. (C) GSE14520 testing cohort. (D) ICGC testing cohort.

training cohort, which was higher than other clinical characteristics, including age (0.512, 0.536, 0.540), gender (0.504, 0.535, 0.565), grade (0.522, 0.505, 0.527), stage (0.659, 0.688, 0.669). Besides, we found the gene signature also performed more specific and sensitive than any single gene (Table 2). The OS in the training cohort was significantly different between low and high risk group, in the 1-year, 3-year and 5-year, the OS in the high risk group were 0.576, 0.181, 0.076, the OS in the low risk group were 0.814, 0.306, 0.142, the result indicated that patients with a high-risk score have more poor OS than the patients with low-risk score ($P < 0.001$), detail was presented in Fig. 4E. By conducting Univariate and

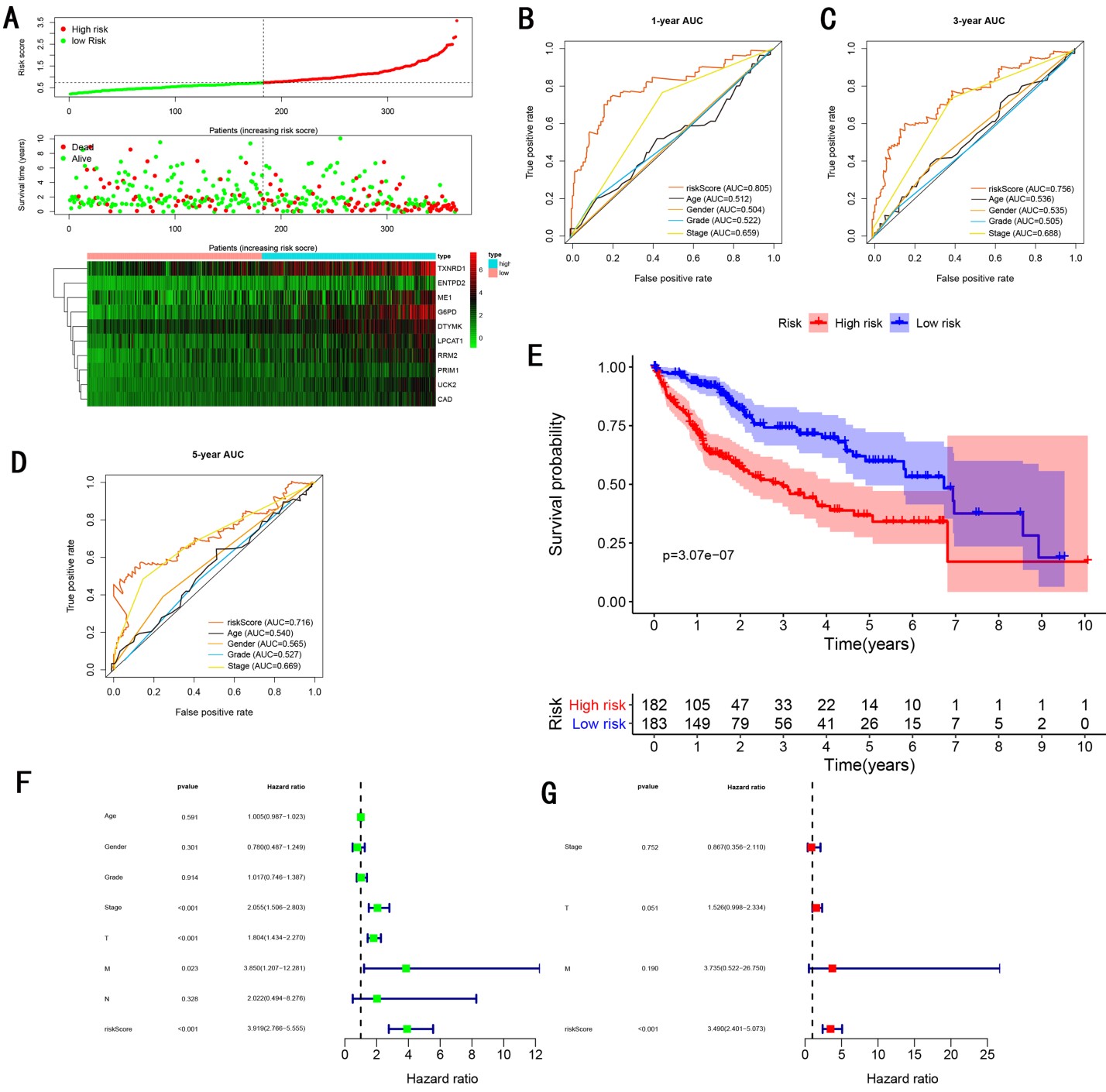

**Figure 4 Gene signature performance analysis using training cohort.** (A) Distribution of 10-gene-based risk scores, patient survival durations, gene expression levels. (B) One-year ROC curve analyses of gene signature and clinical parameters. (C) Three-year ROC curve analyses of gene signature and clinical parameters. (D) Five-year ROC curve analyses of gene signature and clinical parameters. (E) Kaplan–Meier curves of OS based on gene signature. (F) Prognostic value detection of the gene signature via univariate survival-related analysis. (G) Prognostic value detection of the gene signature via multivariate survival-related analysis.

**Table 2 Comparison of the AUC between gene signature and single gene.**

| Characteristic | Training cohort | Internal testing cohort | GSE14520 testing cohort | ICGC testing cohort | Entire testing cohort |
|---|---|---|---|---|---|
| Risk score | 0.786 | 0.773 | 0.707 | 0.775 | 0.732 |
| G6PD | 0.738 | 0.741 | 0.575 | 0.698 | 0.665 |
| LPCAT1 | 0.708 | 0.680 | 0.514 | 0.722 | 0.663 |
| ME1 | 0.630 | 0.642 | 0.514 | 0.597 | 0.586 |
| PRIM1 | 0.706 | 0.748 | 0.534 | 0.707 | 0.634 |
| RRM2 | 0.716 | 0.754 | 0.501 | 0.698 | 0.655 |
| TXNRD1 | 0.674 | 0.663 | 0.532 | 0.611 | 0.599 |
| UCK2 | 0.736 | 0.699 | 0.650 | 0.711 | 0.695 |
| CAD | 0.743 | 0.726 | 0.602 | 0.688 | 0.676 |
| DTYMK | 0.689 | 0.722 | 0.534 | 0.734 | 0.641 |
| ENTPD2 | 0.552 | 0.546 | 0.503 | 0.662 | 0.580 |

Multivariate Cox regression analysis, we noted that gene signature have a significant correlation with worse OS, the HR of the gene signature was 3.919 (95% CI [2.766–5.555]) with $P$-value < 0.001 in Univariate Cox regression analysis (Fig. 4F), 3.490 (95% CI [2.401–5.073]) with $P$-value < 0.001 in Multivariate Cox regression analysis (Fig. 4G), thus the gene signature was an independent prognostic factor of other clinical variables.

### Validation of the gene signature

To validate the predictive ability in different HCC populations, we applied the gene signature to ICGC testing cohort, the result was similar to the training cohort. Figure 5A showed the distribution of risk scores for each patients, patients in high risk group had a worse OS than patients with a low-risk group. In addition, the AUC for 1-year (Fig. 5B), 3-year (Fig. 5C), and 5-year (Fig. 5D) OS were 0.775, 0.754, 0.778, which was higher than other clinical characteristics, including age (0.542, 0.520, 0.595), gender (0.587, 0.574, 0.545), prior Maliganancy (0.526, 0.572, 0.508). Even though the AUC of gene signature was a little less than the TNM stage at 1-year OS (0.775 vs 0.809), the AUC of gene signature was much larger than the TNM stage at 3-year OS (0.754 vs 0.658), 5-year OS (0.778 vs 0.564). The AUC of ROC for gene signature was obviously greater than single gene (Table 2). Moreover, the OS for patients in the high risk group was 0.576 at 1-year, 0.181 at 3-year, 0.076 at 5-year, compared with 0.814, 0.306, 0.142 in the low risk group ($P$ < 0.001, Fig. 5E). Further Univariate Cox regression analysis and Multivariate Cox regression analysis displayed gene signature was a powerful and independent factor in external testing cohort (Figs. 5F and 5G). For GSE14520 testing cohort (Fig. 6), internal testing cohort (Fig. 7) and entire testing cohort (Fig. 8), the gene signature had the similar predictive ability. The distribution of risk scores, gene expression were evaluated in GSE14520 testing cohort (Fig. 6A), internal testing cohort (Fig. 7A) and entire testing cohort (Fig. 8A). The ROC curve demonstrated that gene signature was more specific and sensitive than any clinical characteristics and any single gene in GSE14520 testing cohort

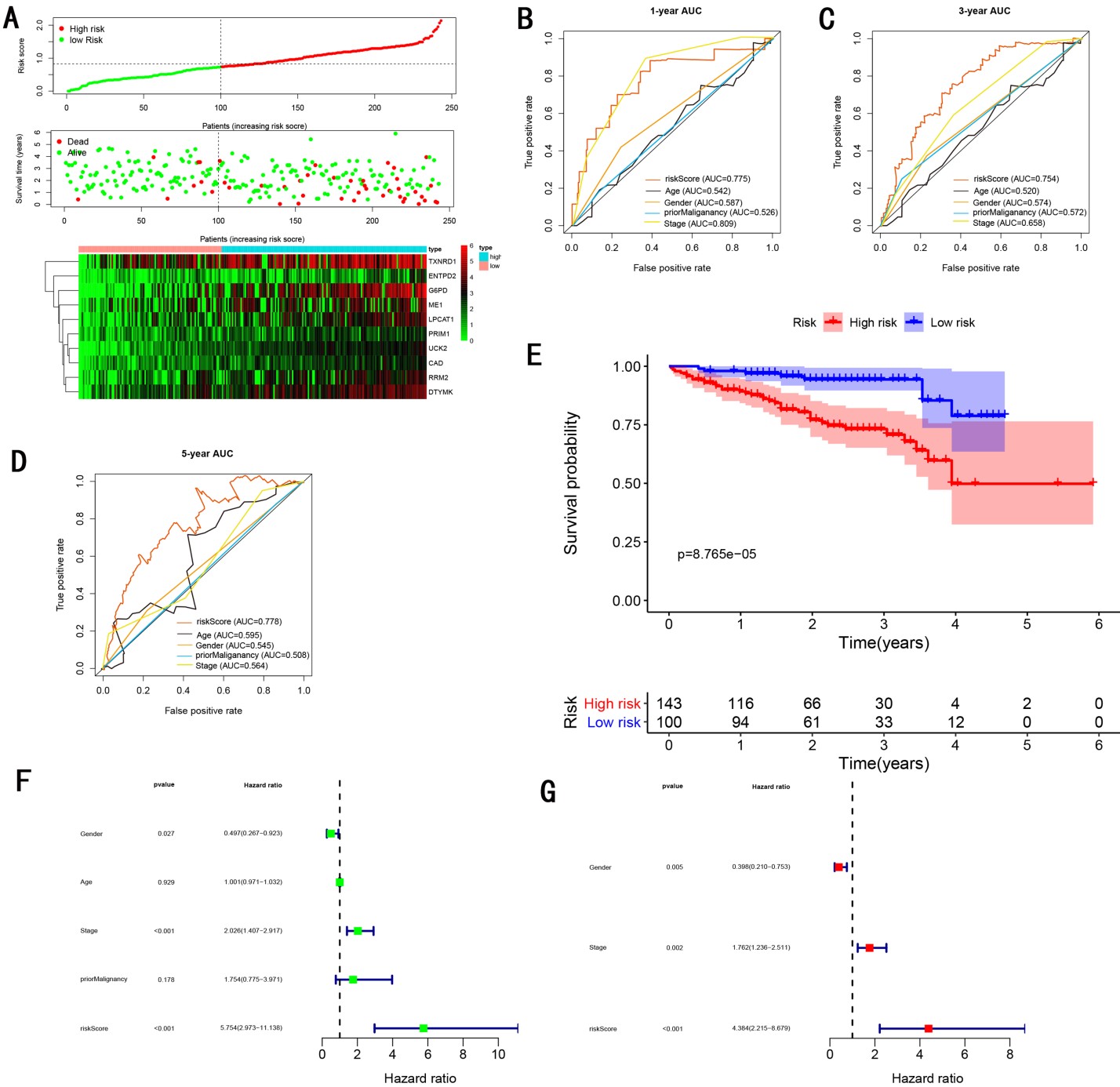

**Figure 5 Gene signature performance analysis using ICGC testing cohort.** (A) Distribution of 10-gene-based risk scores, patient survival durations, gene expression levels. (B) One-year ROC curve analyses of gene signature and clinical parameters. (C) Three-year ROC curve analyses of gene signature and clinical parameters. (D) Five-year ROC curve analyses of gene signature and clinical parameters. (E) Kaplan–Meier curves of OS based on gene signature. (F) Prognostic value detection of the gene signature via univariate survival-related analysis. (G) Prognostic value detection of the gene signature via multivariate survival-related analysis.

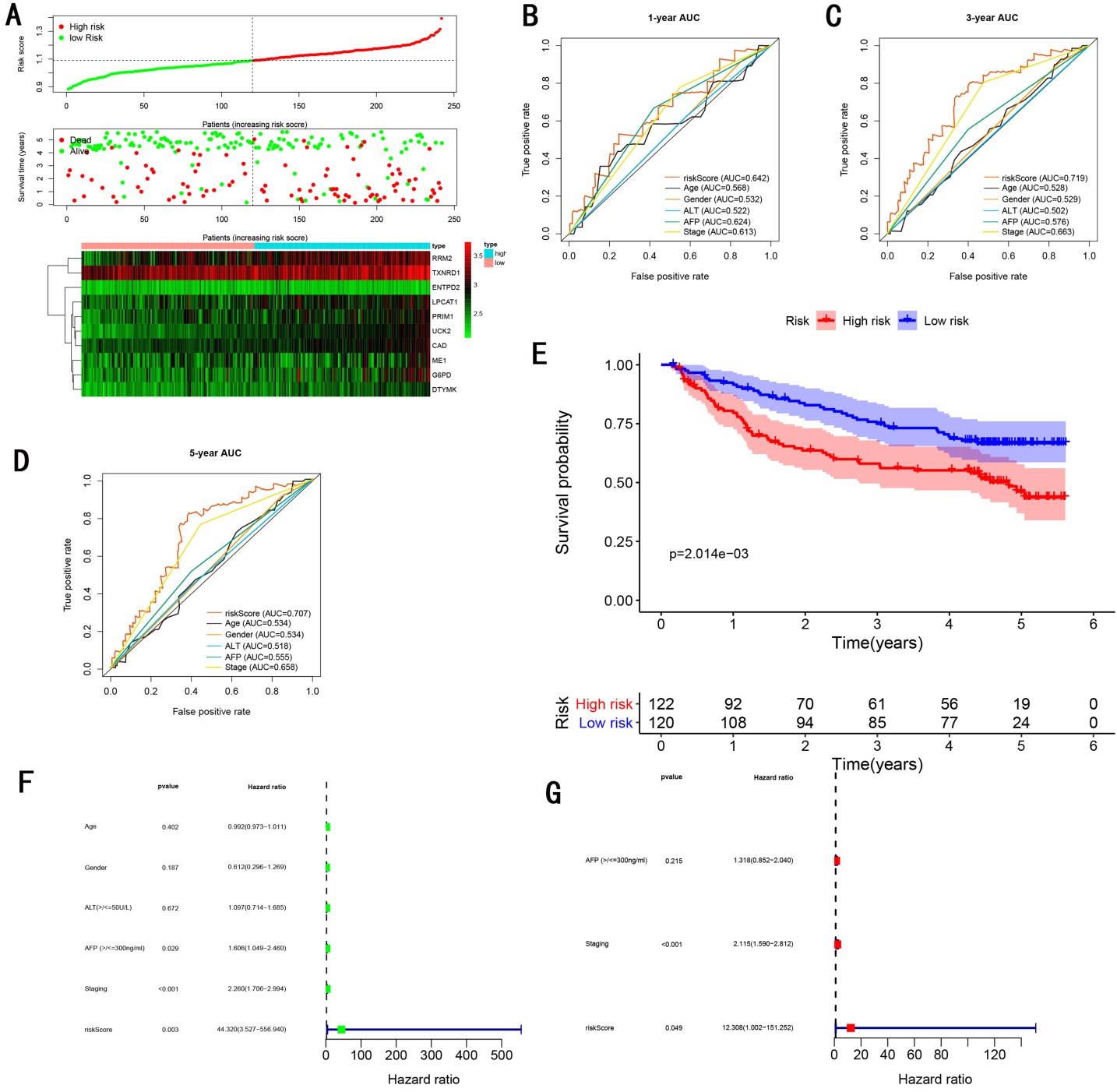

**Figure 6 Gene signature performance analysis using GSE14520 testing cohort.** (A) Distribution of 10-gene-based risk scores, patient survival durations, gene expression levels. (B) One-year ROC curve analyses of gene signature and clinical parameters. (C) Three-year ROC curve analyses of gene signature and clinical parameters. (D) Five-year ROC curve analyses of gene signature and clinical parameters. (E) Kaplan–Meier curves of OS based on gene signature. (F) Pognostic value detection of the gene signature via univariate survival-related analysis. (G) Prognostic value detection of the gene signature via multivariate survival-related analysis.

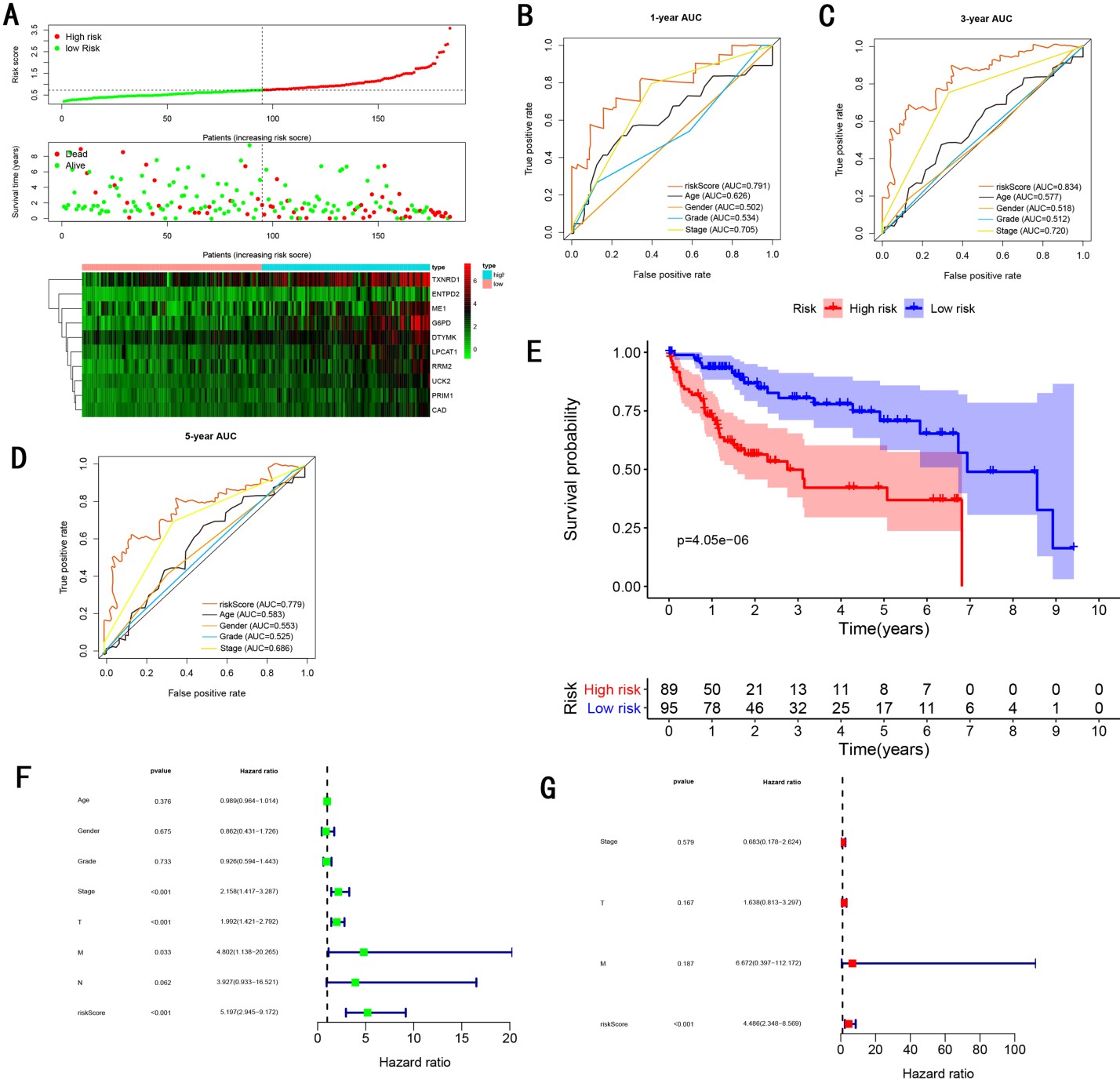

**Figure 7 Gene signature performance analysis using internal testing cohort.** (A) Distribution of 10-gene-based risk scores, patient survival durations, gene expression levels. (B) One-year ROC curve analyses of gene signature and clinical parameters. (C) Three-year ROC curve analyses of gene signature and clinical parameters. (D) Five-year ROC curve analyses of gene signature and clinical parameters. (E) Kaplan–Meier curves of OS based on gene signature. (F) Prognostic value detection of the gene signature via univariate survival-related analysis. (G) Prognostic value detection of the gene signature via multivariate survival-related analysis.

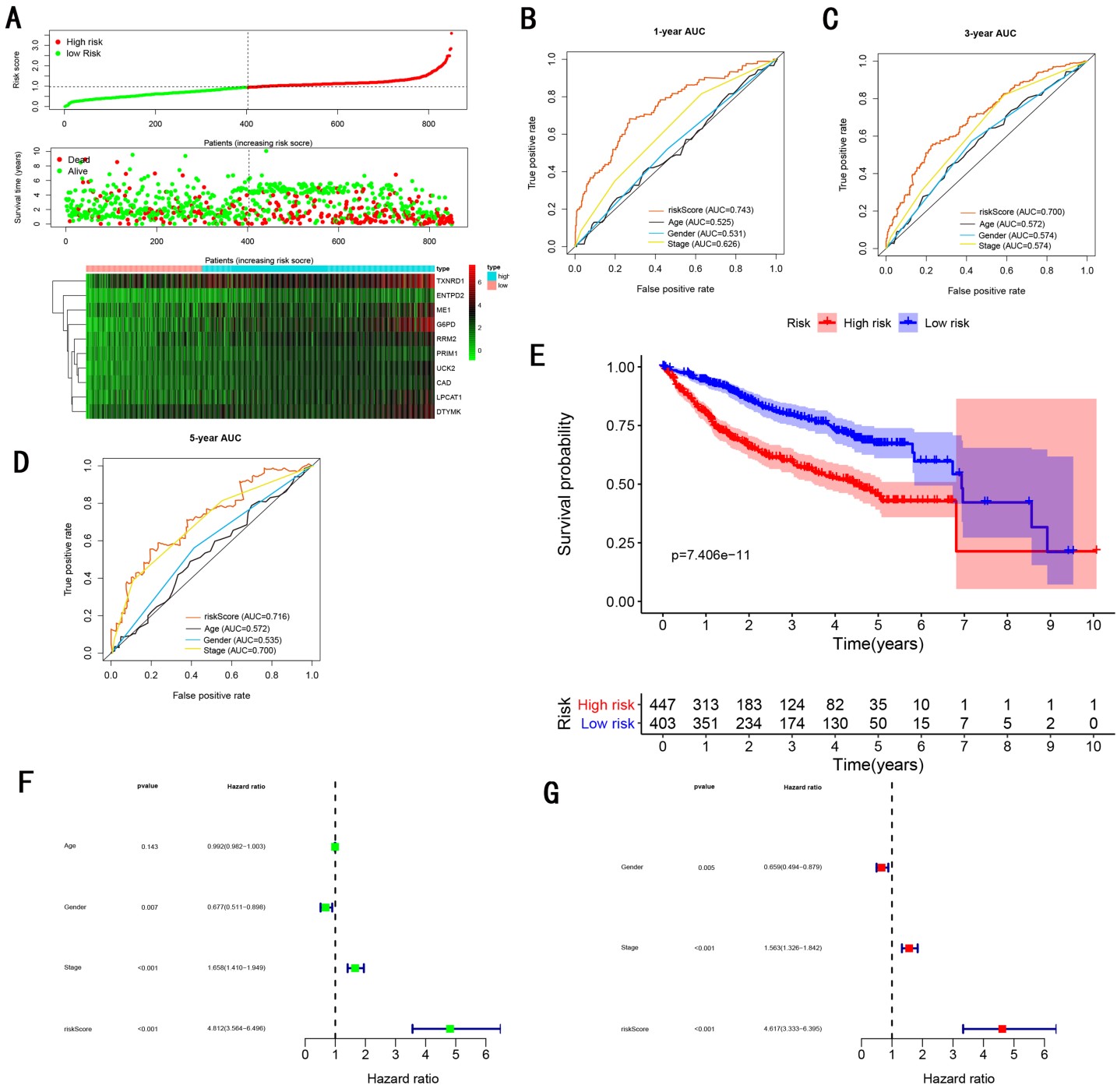

**Figure 8 Gene signature performance analysis using entire testing cohort.** (A) Distribution of 10-gene-based risk scores, patient survival durations, gene expression levels. (B) One-year ROC curve analyses of gene signature and clinical parameters. (C) Three-year ROC curve analyses of gene signature and clinical parameters. (D) Five-year ROC curve analyses of gene signature and clinical parameters. (E) Kaplan–Meier curves of OS based on gene signature. (F) Prognostic value detection of the gene signature via univariate survival-related analysis. (G) Prognostic value detection of the gene signature via multivariate survival-related analysis.

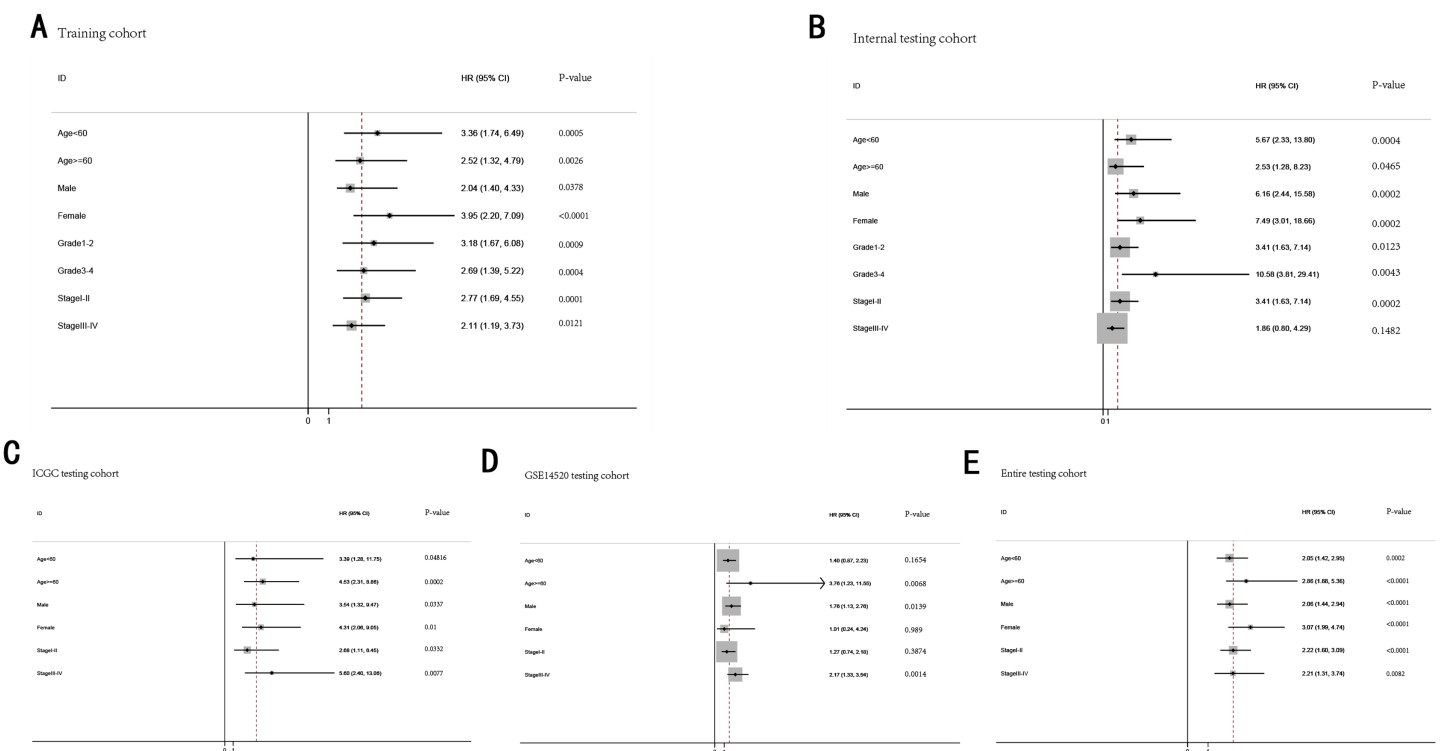

**Figure 9 The predictive performance of the gene signature on OS in different subgroups stratified by clinical parameters.** (A) Training cohort. (B) Internal testing cohort. (C) ICGC testing cohort. (D) GSE14520 testing cohort. (E) Entire testing cohort.

(Figs. 6B–6D; Table 2), internal testing cohort (Figs. 7B–7D; Table 2) and entire testing cohort (Figs. 8B–8D; Table 2). Patients in a high-risk group have poorer OS than the patients in low-risk group for GSE14520 testing cohort (Fig. 6E), internal testing cohort (Fig. 7E), entire testing cohort (Fig. 8E) (P < 0.001). Univariate and Multivariate Cox regression analysis indicated the gene signature was an independent prognostic factor for GSE14520 testing cohort (Figs. 6F and 6G), internal testing cohort (Figs. 7F and 7G) and entire testing cohort (Figs. 8F and 8G).

## Stratification analysis

To further demonstrate the clinical significance of the gene signature in HCC, we perform the survival analysis stratified by clinical variables (age, gender, grade, stage) in training cohort and internal testing cohort, by clinical covariates (age, gender, stage) in ICGC testing cohort, GSE14520 testing cohort and entire testing cohort. Patients of stage I–II, stage III–IV, grade 1–2, grade 3–4, age <60, age >=60, female, male were stratified into high risk group and low risk group. The log-rank test indicated that HCC patients in high risk group still had obviously worse OS than patients in low risk group for training cohort (Fig. 9A), internal testing cohort (Fig. 9B), ICGC testing cohort (Fig. 9C), GSE14520 testing cohort (Fig. 9D) and entire testing cohort (Fig. 9E), and the high-risk patients of subgroup subdivided by the signature had poorer survival than the low-risk patients in training cohort (Figs. S1A–S1H), internal testing cohort (Figs. S1I–S1P), GSE14520 testing

cohort (Figs. S2A–S2C, S2E and S2F), ICGC testing cohort (Figs. S2G–S2L) and entire testing cohort (Figs. S3A–S3F). There was no different trend between high and low risk group for female patients in GSE14520 testing cohort, small number female patients may be an important reason. Interestingly, subgroup stage I, stage II, stage III in entire testing cohort were stratified into high risk group and low risk group, HCC patients in high risk group had poorer OS than patients in low risk group (Figs. S3G–S3I).

Bidkhori et al. (2018) published a article that was proposing 3 sub-types of HCC, including iHCC1, iHCC2 and iHCC3. They reported that the expression of iHCC3 tumors are markedly distinct from those of iHCC2 and iHCC1, and a larger number of genes are differentially expressed between iHCC3 and iHCC1/iHCC2 compared with iHCC1 vs iHCC2. Consistent with the result, we revealed that the expression level of majority of genes of the gene signature were higher in iHCC3 than iHCC1/iHCC2, and the expression level of iHCC1 was similar to iHCC2 (Fig. S4).

### Diagnostic capability

The gene signature have been validated the predictive ability in different HCC populations. To further explore diagnostic capability of the gene signature, we compared risk score between normal liver and HCC in training cohort (Fig. S5A, $P < 0.0001$), ICGC testing cohort (Fig. S5B, $P < 0.0001$) and GSE14520 testing cohort (Fig. S5C, $P < 0.0001$), the risk score in HCC was obviously higher in than normal liver. The AUC of ROC curve was 0.98 in training cohort (Fig. S6A), 0.77 in ICGC testing cohort (Fig. S6B) and 0.97 in GSE14520 testing cohort (Fig. S6C), which revealed the strong diagnostic capability for HCC. In addition, we also investigated the distribution of high-risk and low-risk patients in different stages and grades, the proportion of high risk patients is higher in advanced tumor grade (grade 3–4 ) than early tumor grade (grade 1–2) in training cohort (Fig. S5D). The proportion of high risk patients is higher in late stage (TNM III–IV) than early stage (TNM I–II) in training cohort (Fig. S5E), ICGC testing cohort (Fig. S5F) and GSE14520 testing cohort (Fig. S5G). Furthermore, as the TNM stage and tumor grade increased, the risk score increased (Figs. S5H–S5K). For early and advanced tumor grade in training cohort (Figs. S6D and S6E), early and advanced TNM stage in training cohort (Figs. S6F and S6G), ICGC testing cohort (Figs. S6H and S6I), GSE14520 testing cohort (Figs. S6J and S6K), the AUC of ROC curve indicated the modest diagnostic capability.

### Comparing the performance of the gene signature with other gene signatures and TMN stage

TMN stage is still significant to predict the survival of HCC patients. In the training cohort, we found the AUC of gene signature was larger than the TNM stage at 1-year, 3-year, 5-year (0.805 vs 0.659 at 1-year, 0.756 vs 0.688 at 3-year, 0.716 vs 0.669 at 5-year ) (Figs. 4B–4D), the result was consistent with GSE14520 testing cohort (Figs. 6B–6D), internal testing cohort (Figs. 7B–7D) and entire testing cohort (Figs. 8B–8D). For ICGC testing cohort (Figs. 5B–5D), even though the AUC of gene signature was a little less than the TNM stage at 1-year OS (0.775 vs 0.809), the AUC of gene signature was much larger than the TNM stage at 3-year OS (0.754 vs 0.658), 5-year OS (0.778 vs 0.564).

Thus we believed the gene signature was more specific and sensitive than TNM stage. Next, we also compared the AUC of ROC between the gene signature and all single genes, the AUC of the gene signature was larger than any single gene (Table 2). Moreover, we further compared the gene signature with other gene signatures, the ROC analysis indicated our model had better performance, 0.805 at 1-year, 0.756 at 3-year, 0.716 at 5-year, the AUC of *Long et al. (2018)* model is 0. 7674, 0.7040, 0.6919 at 1, 3 and 5-year, the AUC of *Qiao et al. (2019)* model is 0.71, 0.69 at 3 and 5-year, the AUC of *Li et al. (2017)* model is 0.67, 0.67 at 3 and 5-year, the AUC of Xiao-Hong *Xiang et al. (2019)* model is 0.708, 0.699, 0.678 at 1, 3 and 5-year, the AUC of *Li et al. (2017)* model is 0.727, 0.709, 0.604 at 1, 3 and 5-year, the AUC of *Yan et al. (2019)* model is 0.712, 0.661 at 1, 3-year, indicating that the model had a high sensitivity and specificity for predict the survival.

### Go enrichment analysis and Gene set enrichment analysis

We performed a differential expression analysis between low and high risk group to reveal the association between the metabolic genes and two subtypes. We identified 18 upregulated and 14 downregulated genes between patients in the low vs high risk group. Go enrichment analysis revealed that upregulated genes in high risk group were mainly involved in cell cycle regulation, such as DNA replication, chromatid/chromosome segregation and regulation, spindle organization and recombination. Upregulated genes in low risk group were mainly involved metabolic and energy regulation, including metabolism of amino acids, oxidative phosphorylation, fatty acid β oxidation and catabolism (Fig. S7A).

In order to further explore the mechanism of prognostic genes in patients with hepatocellular carcinoma, we conducted GSEA between low and high risk group to identify the significant pathways (FDR < 0.05, NOM *P*-value < 0.05). The most meaningful pathway were identified according to the FDR standard, we uncovered the most meaning pathways which were active in the high-risk group, including KEGG_CELL_CYCLE, KEGG_NUCLEOTIDE_EXCISION_REPAIR, KEGG_OOCYTE_MEIOSIS, KEGG_ PURINE_METABOLISM, KEGG_PYRIMIDINE_METABOLISM. And the most meaning pathways which were active in the low-risk group, including KEGG_DRUG_ METABOLISM_CYTOCHROME_P450, KEGG_FATTY_ACID_METABOLISM, KEGG_GLYCINE_SERINE_AND_THREONINE_METABOLISM, KEGG_PRIMARY_ BILE_ACID_BIOSYNTHESIS, KEGG_VALINE_LEUCINE_AND_ISOLEUCINE_ DEGRADATION. The result was consistent with go enrichment analysis, significant pathways which were active in high risk group were mainly related with cell cycle regulation. In contrast, meaningful pathways which were active in low risk group were mainly associated with metabolic and energy regulation (Fig. S7B).

### DISCUSSION

Due to the complicated molecular and cellular heterogeneity in HCC, the morbidity and mortality still remains high level in the world. Novel prognostic biomarkers to predict the survival of HCC patients is urgently needed. Metabolism reprograming is considered the novel hallmark of cancer in the future. However, the number of novel metabolic

biomarkers and prognostic models could be applied to predict the survival of HCC patients is still small. In this study, we constructed a metabolic gene signature by performing training cohort from TCGA dataset, the gene signature showed a strong prognostic performance for predicting the survival of HCC patients. Compared with previous studies and TNM stage, the model possessed a higher sensitivity and specificity. Meantime, the gene signature was an independent prognostic factor of other clinical variables, which showed a high value of HR. The prognostic value of the gene signature was validated by performing internal testing cohort, ICGC testing cohort, GSE14520 testing cohort and entire testing cohort. Furthermore, the gene signature demonstrated a high diagnostic capability in differentiating normal tissue and HCC, and showed a modest diagnostic capability in early and advanced TNM stage, early and advanced grade. Finally, several significant biological processes signaling pathways underlying hepatocellular carcinoma were identified for further validation.

We identified ten risky prognostic genes *(CAD, DTYMK, ENTPD2, G6PD, ME1, RRM2, TXNRD1, UCK2, LPCAT1, PRIM1)*. *CDA* gene encodes a 243-kDa multifunctional protein, consists of carbamyl phosphate synthetase (CPSase), aspartate transcarbamylase (ATCase), glutamine amidotransferase (GLNase), dihydroorotase (DHOase) (*Kim, Kelly & Evans, 1992*). *CDA* is the main participant in de novo pyrimidine synthesis, which is very important to provide malignant cells and proliferating cells with nucleotides for DNA replication (*Aoki & Weber, 1981*; *Fairbanks et al., 1995*). Therefore, the upregulation of *CAD* may be considered as a prognosis biomarker and therapeutic target. *Uhlen et al. (2015)* have reported that the expression level of *CAD* is high, *Morin et al. (2012)* indicated the expression level of *CAD* was associated with local tumor extension and cancer relapse and identified *CAD* as a potential predictive marker of cancer relapse, *Sigoillot, Sigoillot & Guy (2004)* have showed the intracellular *CAD* concentration was 3.5- to 4-fold higher in MCF7 cells than that in normal MCF10A breast cells, and MAP kinase activity and a nonclassical ERalpha/Sp1-mediated pathway may account for the high *CAD* level (*Khan et al., 2003*). Previous study have indicated the expression level of *CAD* is higher in hepatoma carcinoma cell than normal liver cell, however, The prognostic value of *CAD* for hepatocellular carcinoma has not been validated. *DTYMK* is a nuclear-encoded deoxythymidylate kinase, which expressed in all tissues and participate in the activity of dTTP production (*Caspi et al., 2016*), and is a key part for DNA synthesis. *Liu et al. (2013b)* have indicated the expression of *DTYMK* is increased in lung adenocarcinomas in comparison to normal lung, and identified elevated *DTYMK* expression as an unfavorable predictor. *Yeh et al. (2017)* found the *DTYMK* was a poor prognostic factor in HCC. *DTYMK* was observed in the 5-FU resistant colon cancer cells, which may provide a new therapy for the HCC by applying the 5-FU combination therapy. *ENTPD2* belongs to ENTPD family (*Chiu et al., 2017*), have reported only ENTPD1 plays an important role in cancer, however, discloses that *ENTPD2* is harnessed by cancer cells to escape immune-mediated destruction and the expression level of *ENTPD2* was also high in HCC patients, and the high expression of *ENTPD2* was associated with direct liver invasion, tumor microsatellite formation and venous invasion, as well as the absence of tumor encapsulation. The pentose phosphate pathway belongs to major carbohydrate pathways,

it could produce ribose and NADPH to protect and promote cells proliferation in hypoxic conditions, such characteristic meets the demand of malignant proliferating cells, therefor, the change of the pentose phosphate pathway may be the landmark of cancer (*Sacoman et al., 2012*). *G6PD* is the rate-controlling enzyme of pentose phosphate pathway, previous researches have reported *G6PD* gene is an oncogene and the expression level upregulates in bladder cancer (*Ohl et al., 2006*), ESCC (*Wang et al., 2016*), breast cancer (*Pu et al., 2015*), *Sun et al. (2014)* indicated *G6PD*-deficient women have reduced breast cancer risk, *Zhang et al. (2017)* indicated overexpression of *G6PD* increases the risk of colon cancer, *Wang et al. (2012a)* reported *G6PD* could promote the progression of gastric cancer cells and is associated with poor clinical outcome for patients with gastric cancer. *Munemoto et al. (2019)* activated *G6PD* gene to accelerates carcinogenesis and cancer progression. Previous studies have detected the *G6PD* is overexpressed in HCC (*Li et al., 2012*; *Xu et al., 2014*), and *Gao et al. (2017)* found *G6PD* WAS significantly changed by using sequential window acquisition of all theoretical mass spectra (SWATH-MS), Hu H et al found the cell migration and invasion ability decreased when the expression of *G6PD* was downregulated. miR122 and miR-1 suppress the expression of *G6PD* to inhibit tumor growth through inhibiting the activity of PPP in hepatocellular cancer (*Barajas et al., 2018*). *Zhao et al. (2018)* found *G6PD* promotes migration and invasion of hepatocellular carcinoma cells through inducing epithelial-mesenchymal transition by activating of transcription 3 (STAT3) pathway. *ME1* is multifunctional protein, which relates glycolytic and citric acid cycles. *ME1* plays an important role in tumor development, it have reported that the expression level of *ME1* is high in various cancers and promotes growth and metastasis, including colorectal cancer, 39 breast cancer (*Liao et al., 2018*), bladder cancer (*Liu et al., 2018*), gastric cancer (*Lu et al., 2018*), nasopharyngeal carcinoma (*Zheng et al., 2012*). Several studies indicates that *ME1* is associated with poor prognosis in hepatocellular carcinomas and OSCC (*Knoblich et al., 2014*; *Wen et al., 2015*), *ME1* promotes HCC metastasis through influencing epithelial-mesenchymal transition (EMT) processes (*Wen et al., 2015*), and *ME1* also could reduce the sensibility of radiation (*Chakrabarti, 2015*; *Woo et al., 2016*). As a vital subunit of rate-limiting catalyzes (ribonucleotide reductase) (RNR), which is necessary for DNA replication and DNA damage repair (*Aye et al., 2015*). *RRM2* was reported to be associated with various cancers, including ovarian cancer (*Wang et al., 2012b*), bladder cancer (*Morikawa et al., 2010b*), colorectal cancers (*Liu et al., 2013a*; *Lu et al., 2012*), gastric cancer (*Morikawa et al., 2010a*). Meantime, several showed up-regulated *RRM2* promotes tumorigenesis, proliferation, and inhibits apoptosis, and is associated with poor prognosis (*Das et al., 2019*; *Kolberg et al., 2017*; *Liang et al., 2019*; *Liu et al., 2013a*; *Souglakos et al., 2008*; *Wang et al., 2012b*). It is known that *RRM2* promotes drug resistance in various cancers (*Goan et al., 1999*; *Nakano et al., 2007*; *Shah et al., 2014*). Thus *RRM2* is consider a novel drug target (*Aye et al., 2015*; *Minami et al., 2015*) including Trans-4,4′-Dihydroxystilbene (*Chen et al., 2019*), COH29 (*Chen et al., 2015*), GW8510 (*Hsieh et al., 2016*). For hepatocellular carcinoma, many bioinformatics analysis indicated *RRM2* is a clinical prognostic markers (*Dawany, Dampier & Tozeren, 2011*; *He et al., 2017*; *Wu et al., 2019*). *Lee et al. (2014)* showed the expression level of *RRM2* is up-regulated and *RRM2* is a significant marker for predicting

clinical prognosis. Several drug may target *RRM2* to suppress hepatocellular carcinoma cells (*Gao et al., 2013*; *Kosakowska-Cholody et al., 2009*). Cancerous cells have to face the increased oxidative stress due to the high metabolism and metabolic disorders, the activation of glutathione (GSH) and thioredoxin (TXN) systems could compensate the severe stress (*Arner & Holmgren, 2006*), thus cancerous cells overactivated GSH and TXN systems to adapt to the oxidative stress, the cytosolic TXN reductase 1 (*TXNRD1*) is a vital part of thioredoxin (TXN) system, which is up-regulated in various cancers (*Cadenas et al., 2010*; *Hughes et al., 2018*; *Lincoln et al., 2003*) and high expression is associated with poor clinical prognosis in multiply types of cancers (*Bhatia et al., 2016*; *Leone et al., 2017*). In addition, *TXNRD1* is also consider as a drug target with high efficacy and low toxicity. Recently, several studies found that the expression of *TXNRD1* is high (*Cadenas et al., 2010*; *Cho et al., 2019*; *Fu et al., 2017*) and is related with poorer clinicopathological features, meantime, inhibition of *TXNRD1* Inhibits the development and progression of hepatocellular carcinoma cells (*Lee et al., 2019a*). *UCK2* gene encodes uridine-cytidine kinase 2, which plays vital role in biosynthesis of the pyrimidine nucleotide (*Schumacher et al., 2013*; *Tomoike et al., 2017*). It have reported that *UCK2* is overexpressed in multiple type of cancers and is associated with poor prognosis, including breast cancer, pancreatic cancer, colon cancer. *Yu et al. (2019)* showed that high *UCK2* expression is associated with clinicopathologic feature and is a independent marker for predicting OS and RFS in hepatocellular carcinoma, the expression level may be influenced by the methylation of status cg0927774. In addition, knockdown of *UCK2* suppressed proliferation, migration and invasion (*Huang et al., 2019*), *UCK2* may promotes HCC cell progress through stat3 signaling pathway. The alterations of membrane phospholipid levels could influence membrane fluidity and facilitate metastases because they affect motility, basement membrane invasion, and adhesion (*Taraboletti et al., 1989*). *LPCAT1* is a cytosolic enzyme that converts lysophosphatidylcholine (LPC) to phosphatidylcholine (PC). As an important subtype belongs to the 4 *LPCAT* subtypes (*Shindou & Shimizu, 2009*), *LPCAT1* has been obtained much attention for cancers, *LPCAT1* could contribute the progression, metastasis, and recurrence of cancer. To date, *LPCAT1* over-expression have been reported in multiple types of cancers, including cell renal cell carcinoma (*Du et al., 2017*), gastric cancer (*Uehara et al., 2016*), breast cancer (*Abdelzaher & Mostafa, 2015*), oral squamous cell carcinoma (*Shida-Sakazume et al., 2015*), hepatocellular carcinoma (*Morita et al., 2013*). The DNA primase polypeptide 1 (*PRIM1*) is responsible for synthesizing small RNA primers for Okazaki fragments generated during discontinuous DNA replication, the DNA replication cannot proceed without the catalytic function of *PRIM1*, thus *PRIM1* is an vital role in the initiation (priming) of the DNA replication, and its aberrations may play a key tumorigenic factor by affecting the cell cycle transition from G1 to S phase (*Yotov et al., 1999*). So far, *PRIM1* have been reported be associated with the formation of breast cancer (*Lee et al., 2019b*).

Several significant biological process and signaling pathway which were active in high risk or low risk group have been revealed to provide a new insights of the development of HCC. We have divided patients from training cohort into two subtypes by performing our gene signature. Next, we conducted differential expression analysis between high and

low group to the association between the metabolic genes and two subtypes. We found that upregulated genes in high risk group were mainly involved in cell cycle regulation, and upregulated genes in low risk group were mainly involved metabolic and energy regulation. Likely, GSEA analysis have demonstrated the similar result. Significant pathways which were active in high risk group were mainly related with cell cycle regulation. In contrast, meaningful pathways which were active in low risk group were mainly associated with metabolic and energy regulation.

## CONCLUSION

The study identified ten prognostic genes that participate aberrant metabolism in HCC, and developed a metabolic ten-gene signature which provides more powerful prognostic information and improve the survival prediction for HCC. We could also score each patient according to the metabolic nine-gene signature and thus identified high-risk HCC patients. The prognostic ability of ten-gene signature was validated by internal testing cohort, GSE14520 testing cohort, ICGC testing cohort, entire testing cohort. Moreover, several significant biological process and signaling pathway have been identified to provide a new insights of the development of HCC. However, further biological experiments should performed to validate our results.

## ACKNOWLEDGEMENTS

We are grateful to the reviewers for their constructive comments which led to improvements in this manuscript. In addition, thanks to Bin Zhao (Official Wechat Account: Bio_Med2017) of Xiamen University and Hexin Lin of Fujian Medical University for suggestions on the manuscripts.

### Funding
This work was supported by Xiamen Scientific and Technological Plan (No. 3502Z20194005, 3502Z20184020). The funders had no role in study design, data collection and analysis, decision to publish, or preparation of the manuscript.

### Grant Disclosures
The following grant information was disclosed by the authors:
Xiamen Scientific and Technological Plan: 3502Z20194005, 3502Z20184020.

### Competing Interests
The authors declare that they have no competing interests.

### Author Contributions
- Zhipeng Zhu conceived and designed the experiments, analyzed the data, prepared figures and/or tables, authored or reviewed drafts of the paper, and approved the final draft.

- Lulu Li performed the experiments, authored or reviewed drafts of the paper, and approved the final draft.
- Jiuhua Xu performed the experiments, authored or reviewed drafts of the paper, and approved the final draft.
- Weipeng Ye performed the experiments, prepared figures and/or tables, and approved the final draft.
- Borong Chen analyzed the data, authored or reviewed drafts of the paper, and approved the final draft.
- Junjie Zeng performed the experiments, prepared figures and/or tables, and approved the final draft.
- Zhengjie Huang conceived and designed the experiments, performed the experiments, authored or reviewed drafts of the paper, and approved the final draft.

### Data Availability
The data and code files are available as Supplemental Files.

The raw data is available from TCGA (search terms: HCC), NCBI GEO (GSE14520) and from the ICGC project (https://icgc.org/; specific datasets used: "Liver Cancer - NCC, JP").

### Supplemental Information
Supplemental information for this article can be found online at http://dx.doi.org/10.7717/peerj.9201#supplemental-information.

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
