# Peer review of "Comprehensive analysis reveals a metabolic ten-gene signature in hepatocellular carcinoma"

_PeerJ, doi:10.7717/peerj.9201_

## Round 0.1 · original submission · Major Revisions

Your manuscript has been reviewed and requires modifications prior to making a decision. The comments of the reviewers are included at the bottom of this letter. Reviewers indicated that the introduction, methods and results sections should be substantially improved. Reviewers also recommended extensive English editing. I agree with the evaluation and I would, therefore, request for the manuscript to be revised accordingly. In addition to this, I strongly recommend the authors discussing the following works, which are highly relevant to the field: PMID: 30336489 and PMID: 30662766.

Reviewer 1 ·

Basic reporting

1. I suggest the author improve the Introduction section by making a careful literature review about previous biomarkers associated with metabolism in liver cancer or metabolic classification for liver cancer.
2. Besides comparing performance of different biomarkers, it is better to compare the subtypes classified by your biomarker with subtypes classified by TCGA group or other well-studied metabolic subtypes (e.g. PMID 30606699).

Experimental design

1. Adjusted p <0.05 was used when identifying DEGs. But P<0.05 was used when identifying prognostic genes. Why do not use same significance level (adjusted p <0.05) ?
2. In Figure 4H, why not consider age, gender, grade and N in Multivariate Cox regression analysis? Same problem with Figure 5H, 6H and 7H.
3. Since the risk-score-based biomarker is sensitive to batch effect, the cutoff set in training dataset may not work in independent dataset. It is better to validate this biomarker in more independent datasets if it is possible.
4. The author should provide more details about the metabolic differences between two subtypes if it is claimed as a metabolic biomarker.

Validity of the findings

1. The result “Interestingly, subgroup stage I, stage 215 II, stage III in entire testing cohort were stratified into high risk group and low risk group, HCC patients in high 216 risk group had better OS than patients in low risk group (Figure S2 M-O)” is unreasonable, which is in conflict with the result when the samples were grouped by stage I-III and stage III-IV.

Additional comments

1. I suggest the author send the manuscript to language edit.
2. The resolution is too low in Figure 8.

Reviewer 2 ·

Basic reporting

They wrote the report in an acceptable English language and can be understood in most cases. There are several areas that can be improved, especially on formulating the sentences to make it more concise and easier to understand (Minor) for example "However, ... still lacking" (line 88-89) can be rephrased to make it more understandable, since this is one of the main kick to explain the background. Another example is line 136-137, about the testing cohort, can be rephrased to make it clearer. My suggestion is that the authors can go through the paper one more time and start cutting complex and long sentences.

The background and context given is clear enough. Same with the references. Structures are clear.

Experimental design

Original primary research within the Aims and Scope of the journal: YES

Research question well defined, relevant & meaningful. It is stated how research fills an identified knowledge gap: YES

Rigorous investigation performed to a high technical & ethical standard.: YES

Methods described with sufficient detail & information to replicate.: YES

Methods are clearly defined. Concerns:
1. What kind of normalization that was used in the paper? TPM, FPKM? Raw counts are not acceptable.
2. I would suggest random partitioning to be used for the DEG calculation due to unbalance number of patients and normal. Otherwise, the result can be skewed.

Validity of the findings

The results were validated with an independent validation cohort (also publicly available data). All data are publicly available.

Regarding the result, several concerns:
1. About the testing cohort: The author mentioned 243 patients from external and 184 from the internal testing cohort, how can the total entire testing cohort be 608? The number doesn't seem to match. Except, if they included the training cohort as well, which doesn't seem necessary.
2. Line 179-182: The information is not clear. The summary is that high-risk subjects have poorer OS score. Not the case in 1-year. Comment?
3. Figure 5B, the stage has higher AUC. Comment?

Suggestion: Bidkhori et.al published a paper that was proposing 3 sub-types of HCC (they were using TCGA data as well). The authors can perform a simple analysis to see how their suggested genes perform for stratification of patients in each sub-type, or as simple as checking the expressions in each sub-type. I think this can add more value and validation to the paper.
https://www.pnas.org/content/115/50/E11874.short?rss=1

MINOR suggestions:
1. (MINOR) Figure 2: IMO, volcano plot should be enough for this. And it would be nice as well to mark the 9 proposed genes in both heatmap and volcano plot.
2. (MINOR) Figure 3: We can see that the proposed genes are higher/lower, but adding statistical analysis between high vs low risk (simple, t-test?) can add more power to the subsequent analysis
3. (MINOR) line 181: more poor --> poorer


In the discussion, each gene was discussed separately. Adding one paragraph to summarize all would be nice, can be with the help of the GSEA result (from the last part of results). My reasoning is that we know that each gene has its own function, but multiple genes can also affect a pathway/biological process, which would be interesting as well to be discussed. For now, the discussion is missing a connecting thread.

Additional comments

In this paper, Zhu et.al is proposing nine gene signature for HCC survival prediction. The analyses were done based on publicly available data (2 datasets). In their result, they showed that the nine gene signature performed better than any of the clinical or singular genes in predicting the low vs high-risk HCC patients.

---

## Round 0.2 · accepted · Accept

The authors addressed the reviewers' concerns and substantially improved the content of MS.

So, based on my own assessment as an editor, no further revisions are required and the MS can be accepted in its current form.

Reviewer 1 ·

Basic reporting

The authors have addressed my concerns adequately.

Experimental design

The authors have addressed my concerns adequately.

Validity of the findings

The authors have addressed my concerns adequately.

Additional comments

The authors have addressed my concerns adequately.

Reviewer 2 ·

Basic reporting

The author has performed adequate language editing to an acceptable level.

Experimental design

Please include the use of FPKM value in the method part. Important for reproducibility.

Validity of the findings

No comment

Additional comments

Authors have made the required revision to make it as acceptable